# Comparison of Self-Care Practices and BMI between Celiac and Non-Celiac Adolescent Populations

**DOI:** 10.3390/healthcare12171796

**Published:** 2024-09-08

**Authors:** Montserrat Monserrat-Hernández, Juan Carlos Checa-Olmos, Ángeles Arjona Garrido, Rafael Galera-Martínez

**Affiliations:** 1Department of Geography, History and Humanities, University of Almería, 04120 Almería, Spain; mmh548@ual.es (M.M.-H.); arjona@ual.es (Á.A.G.); 2Pediatric Gastroenterology and Nutrition Unit, Pediatrics Department, Torrecárdenas Hospital, 04009 Almería, Spain; galeramartinez@gmail.com

**Keywords:** celiac disease, celiac population, self-care, adolescents

## Abstract

Celiac disease (CD) is a chronic gastrointestinal disorder that is most frequently found in Western countries, and the only treatment available today is a lifelong gluten-free diet (GFD). The main aim of the present study is to compare the self-care practices and body mass index (BMI) of adolescents with CD and without CD in different dimensions (physical, psychological, social, and management). The non-probabilistic sample included 236 participants (118 with CD) aged between 12 and 16 years old, who were part of the project “Role of the family in the perception of youth self-care”. The data were obtained through the self-administration of the Practice and Management of Youth Self-care questionnaire, while the celiac population also completed the Adherence to a Gluten-Free Diet (GFD) questionnaire. The celiac population analyzed showed significant correlations between the Physical Practices and Management with adherence to a GFD, and between a GFD and Psychological Practices. Differences were observed between the celiac and the non-celiac populations in the dimensions of Physical Practices, Social Practices, and Management. With respect to BMI, age and CD showed a significant influence of a GFD on BMI (*p* < 0.001). In conclusion, the application of multi-dimensional questionnaires and their relationship with the adherence to a GFD provide valuable information to propose interventions directed to this population.

## 1. Introduction

Celiac disease (CD) is a chronic gastrointestinal disorder that is most frequently found in Western countries, with a prevalence between 0.5 and 1% [1] and, according to the Spanish Association of Pediatrics [2], the incidence in most European countries is one per 100 adolescents, reaching three per 100 in some countries. Presently, the only valid treatment is to follow a gluten-free diet GFD) throughout life. A bad adherence can result in an increase in the incidence of associated gastrointestinal and metabolic diseases, such as irritable bowel disease, malignant gastrointestinal neoplasms, nutrient absorption problems, diabetes mellitus, etc., and, especially in infancy and childhood, developmental delay, damage to the tooth enamel, short stature, delayed puberty, and even neurological symptoms such as hyperactivity or learning disabilities [3,4], and in all cases, a decrease in the quality of life of the patient.

Different methods have been described to detect CD, such as immunogenic gluten peptides (GIP) [5,6], which allow for the early detection of small gluten intakes. This method, just as other more traditional ones, highlights the consequences of exposure to gluten, but does not explain the exposure to it nor whether it can be managed by the patient through self-care.

In addition, celiac disease patients have a higher risk of being overweight and obese during the first year after starting the GFD, given that once the diagnosis has been performed, they can eat without symptoms and improve their ability to absorb nutrients [7,8]. Furthermore, the change to a GFD may also result in an increase in blood pressure (BP), body mass index (BMI), waist circumference, blood glucose levels, and serum lipids [9,10,11,12,13].

Thus, understanding the self-care of young people with celiac disease is indispensable to obtain information regarding the dimensions involved in adherence to the GFD in order to develop effective health promotion practices and to know if these guidelines are similar to those of young people of the same age without CD or if there are significant differences that should promote special practices. Self-care is understood as the set of actions performed by a person to control internal or external factors that may compromise their life and posterior development [14,15]. It is believed that a deficit in self-care exists when people are subjected to limitations that incapacitates them from continuous care or makes it so that this care is inefficient or incomplete [16]. In this aspect of the prevention, most of the studies have been centered on analyzing the descriptive behavioral profile centered on the practices that are directly related with the disease, that is, if preventive measures are practiced [17,18,19,20]. Nevertheless, not many works have analyzed the factors that have an influence on specific behaviors and their relationship with the social and psychological well-being of the individuals studied.

As of the present day, the models of management of self-care have been directed towards the analysis of achievements and mistakes committed during process of medical or nursing care [21,22,23]. For this, interdisciplinary teams are created to promote the self-care and promotion of health, especially in a vulnerable or sick population in health centers and/or doctor’s consultations [24,25]. Now, the management models in terms of prevention, and especially in the adolescent age groups, are developed by the school nurse [26,27], who helps the education community (students and staff) to maintain their health, but not in a generalized manner [28].

To measure self-care, with respect to the population considered not sick, questionnaires exist that can be used to assess the level of care with respect to physical, psychological, or social dimensions [29,30,31]. A unified scale did not exist until the creation of the Practice and Management of Youth Self-Care (PGJ-Q) [32]. Also, diverse questionnaires can be found that measure the self-care of different illnesses or pathologies, which are centered on practices, habits, behaviors, and/or common ailments in each of them [33,34].

The international and Spanish literature about the self-care of celiac patients is scarce, and mainly alludes to the changes in feeding habits and the need to create adherence to a GFD [35,36,37,38,39], and only a few studies in the field of psychology, such as those by Guedes et al. [40] and Lee et al. [41], pay attention to the possible anxiety and depression derived from it. Others highlight the dimensions that intervene in those who suffer from the disease, such as personal, family, emotional, and financial dimensions [28,42,43,44].

Other studies analyze the social relationships and the demand from families for a greater social awareness and support in every sphere: educational, communication media, etc. [45]. In particular, the work by Cañameras [46] describes how families adapt their personal networks to their new circumstances, seeking and sometimes obtaining the social support they need. Celiac disease patients who are diagnosed at an early age experience a certain marginalization in their social relations through food, which must be addressed by specialists [47,48]. Even Galego [49] indicates the need to work with families on the emotional impact.

In summary, the present study seeks to provide information on the situation of adolescent celiac patients with respect to their process of self-care and BMI parameters during their process of adoption of a GFD, by comparing them to a population of adolescents without celiac disease, to be able to define specialized analysis guidelines that go beyond the clinical diagnosis, to, therefore, improve their quality of life.

In addition, derived from the main objectives, the main hypotheses of our study are as follows: (1) Celiac youth who follow a GFD will develop more effective self-care practices regarding physical care and management of their disease compared to youth without CD; (2) celiac youth who follow a GFD will develop less effective self-care practices with respect to youth without CD in social and psychological aspects; (3) age and CD diagnosis will have a significantly inverse relationship with self-care practices; and (4) celiac youth who follow a GFD will have higher BMI than youth without CD.

## 2. Materials and Methods

### 2.1. Participants

Official data on celiac disease according to age were obtained from the General Directorate of Health Care and Health Results from the Comprehensive Plans Service (Dirección General de Asistencia Sanitaria y Resultados en Salud Servicio de Planes Integrales), for ages between 12 and 16 years old = 0.98%) [50,51], to obtain a target celiac disease population of 1370 individuals. The sample size was calculated with the Epidat 4.2 program (Epidemiology Service of the Direccion Xeral de Saùde Pùblica da Consellería de Sanidade; Xunta de Galicia, Spain). With a confidence level of 95% and a margin of error of 0.05, a minimum required sample size of 99 participants was calculated. A total of 118 adolescents with CD participated in the study.

The selection of participants was performed with a non-probabilistic sample from education centers in Andalusia, Spain, who were informed about the possibility of participating in the pilot project.

The celiac population was obtained through the screening questions “check if you have suffered or suffer from the following pathologies and/or diseases”. Nevertheless, to increase the size of the sample of celiac disease population, their families and colleagues were included through the “snowball” technique, to obtain a more representative population.

The recruitment of the non-celiac disease group, to compare it with the celiac group, was performed through the randomized sampling of 1280 surveys of participants with CD to obtain 118 participants for the group.

The eligibility criteria for the non-celiac disease population were being between 12 and 16 years old and residing in Andalusia. For the celiac disease population, the criteria also included being between 12 and 16 years old and residing in Andalusia, as well as having an absence of other chronic metabolic or cardiovascular diseases (whether associated with CD or not) and a CD diagnosis for more than a year.

### 2.2. Instrument

To measure sociodemographic factors, an ad hoc questionnaire was created, which included the following variables: sex (male/female), age, weight (kg), height (m), and length of time following a GFD (only for the celiac population).

The Practice and Management of Youth Self-Care (PGJ) questionnaire [32,52] was utilized. It is composed of 60 questions, and it obtained a Cronbach’s Alpha reliability score α =84 and McDonald’s Omega ώ = 72 for our sample population. This instrument is designed to assess self-care practices (physical, psychological, and social) and overall management. Each item is answered with a Likert-type scale of five options (0: never; 1: almost never; 2: occasionally; 3: almost always; 4: always). The results can be observed in each of the dimensions: Physical Practices (risky behaviors, hygiene, physical activity, and rest) (low ≤ 36 ≤ medium ≤ 56 ≤ high), Social Practices (family, friends, partner, and professionals) (low ≤ 53 ≤ medium ≤ 73 ≤ high), Psychological Practices (spirituality, self-esteem, satisfaction with life, and management of emotions) (low ≤ 38 ≤ medium ≤ 52 ≤ high), and Management (unified, in which a question is posed about the management of diseases and/or preoccupation for caring for one’s health) (low ≤ 16 ≤ medium ≤ 26 ≤ high).

The Celiac Dietary Adherence Test (CDA-T) [53] was used to assess the adherence to a GFD of the celiac population (CP). This questionnaire is composed of 7 items distributed into 3 factors: symptomatology (items 1 and 2), motivation and self-efficacy (items 3, 4, 6, and 7), and mood (item 5). All the items are scored with a Likert-type scale. Items 1 and 2 are scored from 1 (at all times) to 5 (never); items 3, 4, 5, and 6 are scored from 1 (completely agree) to 5 (completely disagree). Item 7 is scored from 1 (never) to 5 (more than ten days). The total score ranges from 7 to 35, where scores lower than 13 indicate an excellent or good adherence, and scores above 17 indicate a low adherence. The questionnaire obtained a good internal consistency for our study sample, with a Cronbach’s α = 0.74 and McDonald’s omega ώ = 0.69 [54].

### 2.3. Data Collection

The present study is part of the project “Role of families in the practices of self-care of young Andalusians”, funded by the Andalusian Public Foundation Center for Andalusian Studies (PRY083/22). The protocol was approved by the Bioethics Committee from the University of Almeria (UALBIO2023/051).

The principal investigator contacted school nurses working at the education centers, who informed the parents of potential participants about the possibility of voluntarily participating in the study, by sending an email to the Association of Student’s Parents. Those who were interested were asked to participate in an online meeting, where all the families who decided to participate signed an informed consent about the aims of the study, as well as the uses of the data obtained, verifying anonymity and the possibility of leaving the study whenever they wished. Afterwards, a link was sent for both parents and adolescents to complete, available at the Limesurvey website (https://www.limesurvey.org/es accesed on 31 march 2024). The questionnaire could be completed from December 2023 to March 2024. The questionnaire was composed of (1) information about the study, (2) sociodemographic characteristics of the adolescents and parents and/or legal tutors, and (3) questionnaires of the outcome variables. The estimated time needed to complete the questionnaires was 20 to 30 min.

### 2.4. Statistical Analysis

The data were analyzed with the IBM SPSS Statistics 27.0 software program. In first place, a descriptive analysis was performed of the categorical variables, including frequencies and percentages, while the quantitative variables were calculated through central tendency and dispersion measurements (mean and standard deviation). Two-tailed correlations were established between the dimensions of the PGJ-Q, BMI, CDA-T, and GFD duration. The normality and homogeneity tests (Kolmogorov–Smirnov and Levene’s) for the dependent variables allowed us to apply parametric tests.

Student’s t-tests for independent samples were performed to verify if significant differences were present between the celiac population (CP) and non-celiac population (NCP) groups with respect to the different dimensions of self-care and BMI. Cohen’s D was calculated as a measurement of the effect size, to compare both independent groups. Afterwards, mixed ANOVA tests were performed to verify if there was an influence of having celiac disease and age on each of the dimensions of self-care and BMI. The Bonferroni test was used as a post hoc test.

## 3. Results

The sample extracted for the analysis was composed of 236 individuals (60.9% female, 42.1% male), aged between 12 and 16 years old (M = 14.2; SD = 1.14) living in the Autonomous Community of Andalusia in the Southeast of Spain. Of these, 118 adolescents suffered from celiac disease (79 women and 39 men). Their mean age was 14.75 years (SD = 2.1) and they had followed a GFD for a mean duration of 10.8 years (coinciding with the diagnosis) (SD = 3.98).

### 3.1. Self-Care, BMI, and Adherence to a GFD by the CP

In general, as shown in Table 1, the CP surveyed obtained scores considered to be “low” for the Physical Practice dimension and “medium” for the Management, Social Practice, and Psychological Practice dimensions.

Positive correlations were observed between Physical Practice and Management of self-care, and also between Management and Psychological Practice.

Physical Practice was negatively correlated with Social Practice and with BMI levels, so that lower BMI values resulted in higher scores in the questionnaire.

On its part, BMI was only correlated with the dimension of Physical Practice.

With respect to adherence to a GFD, a positive relation was observed with the dimensions Physical Practice and Management. The time spent following a GFD was positively correlated with the psychological dimension.

### 3.2. Comparison between the CP and NCP in the Practices of Self-Care and BMI

Significant differences were found between the CP and NCP with respect to the scores in the Physical (*p* < 0.001) Practice, Social (*p* < 0.001) Practice, and Management (*p* = 0.016) dimensions. However, significant differences were not found regarding Psychological Practice (*p* = 0.393) (see Table 2).

In the Physical Practice dimension, the differences were considered as having a large effect size (d_cohen_ = 0.84); in the Social Practice and Management dimensions, they were considered moderate (d_cohen_ = 0.792 and 0.660, respectively); and in the Psychological Practice dimension, an insignificant effect was observed (d_cohen_ = 0.087). With respect to BMI, significant differences were observed between the CP and NCP, with a large effect size (d_cohen_ = 3.667).

### 3.3. Influence of Celiac Disease and Age in the Self-Care Practices and BMI of the Adolescents Surveyed

As observed in Table 3, both groups obtained scores below 36 in the Physical Practice dimension (considered low). The mean for the CP was lower than that found for the NCP (practice of exercise, sleep, hygiene, and harmful behaviors). Also, in both groups, a decrease in the score was observed in those older than 14 years of age, with respect to those who were younger.

In the dimension of Psychological Practice (self-esteem, satisfaction with life, and management of emotions and spirituality), the means in both groups were found to be between 38 and 52 points (considered medium). The CP showed higher scores when participants were younger, as opposed to the NCP.

With respect to Social Practice (relationship with the family, friends, partner, and/or close professionals), the mean of the CP was below 53, considered a “low” score, for participants who were both younger and older than 14 years old. In the NCP, the scores were higher than 72 (high) in both age groups.

In the dimension of Management (development of practices of protection against diseases and/or caring for the disease), in general, both groups obtained a mean between 16 and 26 (a medium level score), and there was a significant increase in the mean in those younger than 14 years old, as compared to those older than 14.

#### 3.3.1. Physical Practice

When analyzing the effect of age and celiac disease on the Physical Practice dimension, we observed that the variable with a significant influence was having celiac disease. There was no significant influence on this dimension of age or the celiac disease*age combination (see Table 4).

Figure 1 clearly shows the difference between the CP and the NCP with respect to the levels of Physical Practice in both age groups. The CP group, independently of age, shows significantly lower levels in this dimension.

#### 3.3.2. Psychological Practice

If we analyze the influence of age and having celiac disease on Psychological Practice, it is observed that none of these variables are significant (see Table 5).

Figure 2 shows that no differences exist between the CP and NCP groups, nor between age groups.

#### 3.3.3. Social Practice

Table 6 shows that, independently, having celiac disease or not has an influence on the dimension of Social Practice, although age does not. When these variables are analyzed together, a significant influence on this dimension is not observed.

Figure 3 shows that the CP (independently of age) has lower scores than the NCP. This means that they have worse relations with their family, peers, and/or partner, and/or with close professionals (teachers, doctors, etc.).

#### 3.3.4. Management

When analyzing the influence of the variables of age and having celiac disease on the dimension of Management, it is observed that a significant influence exists on the score in this dimension only as a function of having celiac disease or not (Table 7).

Figure 4 shows the existence of differences with respect to having celiac disease or not, but not with respect to age in both groups. Those with celiac disease are more worried about how their lifestyle has an influence on their health; they prioritize their health and the related practices, seek help when they are not aware of the solutions to improve their health, and are more aware of activities that have a positive effect on their health.

#### 3.3.5. BMI 

With respect to BMI, an influence was only observed with respect to having celiac disease or not, and not according to age (see Table 8).

Figure 5 shows that having celiac disease obtained higher BMI values.

## 4. Discussion and Conclusions

The main results of the present study show that differences exist among the scores found in the PGJ-Q between the CP and NCP in the dimensions of Physical Practice, Social Practice, and Management. Also, the BMI values were different between the groups.

Presently, not many reliable tools are available to assess the self-care of the celiac disease population, and even fewer have a multi-dimensional characteristic. In particular, the PGJ-Q, previously validated in another study, was considered as a reliable tool in the Andalusian population of adolescents aged 12 to 17 years old [32], and the present study has demonstrated that it would be interesting to consider it with respect to the health practices of the CP. In addition, this field opens new lines in the promotion of GFD and health promotion by establishing the need to promote different health education patterns between PC and PCN.

However, although the PGJ is a useful tool for detecting the dimension on which to act, it is also important to analyze the reasons for the scores in the dimension to be worked on (sociodemographic, psychosocial, emotional, associated comorbidities, etc.) and, therefore, when extrapolating to the clinical setting, this instrument should be used as an initial screening tool to subsequently obtain more detailed information.

As the literature shows, guidelines exist for measuring the level of adherence to a GFD: quantitative or qualitative [55,56,57,58,59], clinical, or through self-reported questionnaires [60,61,62,63]. Likewise, it was verified that there is no exact method for evaluating the causes (factors) that could be directly related with the lack of adherence to a GFD. In this respect, the PGF-Q showed a relationship between its score and Gluten-Free Diet Adherence (GFDA), especially in the dimensions of Physical Practice and Management, related to body care and attention to disease prevention. In addition, the results have indicated that celiac disease is an influencing factor on the score of these variables. With respect to Management, the CP obtained scores that were higher than the NCP ones, perhaps due to the need to develop guidelines with respect to protection, given that their illness is mainly influenced by food [49]. However, in the Physical Practice dimension, the NCP obtained higher scores (healthy eating, rest, and physical activity practices). This could be justified, as the GFD does not necessarily have to be a healthy diet; in fact, most of the gluten-free products have a higher concentration of saturated fats and refined sugars and a higher Glycemic Index that those with gluten [64,65]. These data could coincide with high BMI values in the CP, with these data being similar to those of previous studies on this aspect [11,65].

With respect to Social Practice, the CP obtained lower scores than the NCP, and this could be due to the eating difficulties when eating “outside of the home” experienced by the CP [40,41]. These results are in agreement with studies by Leinonen et al. [39], who showed that following a strict GFD may result in social isolation that could affect self-care at the psychological level, factors which, as in the case of the present study, were correlated in the CP. In addition, social support provides individuals with spiritual support that confers a positive social identity. It is not yet clearly established how the most stable characteristic of the individual limits or favors certain types of coping responses, nor the value of personality traits and the social concept in the willingness to respond in a particular way, but what is undeniable is the influence of all these factors on the psychological response. Perhaps these data point to the need to emphasize psychological and social self-care for the adequate general health of the CP, from which physical practices that are considered acceptable but not perfect may derive. The results have shown use that age is not as important as having celiac disease or not, in the scores found in the dimensions Physical Practice, Social Practice, and Management. The group composed of adolescents younger than 14 obtained higher scores in all the self-care dimensions analyzed. Although not significant, these results may be an approximation of the difficulty in controlling the variables described when individuals have greater autonomy and more social practices outside the family environment [66,67].

As for Psychological Practices, our hypothesis expected a lower value in the CP, but this was not the case; this finding was different to the results of other studies performed on the subject, in which it is clear that the subjects studied, in addition to their physical and symptomatological problems, had negative feelings and emotions such as social stigma, fear or concern about health or causing discomfort, perceptions of inequality, and shame or guilt [49,64], which can lead to psychological disorders such as anxiety and depression [68,69]. Also, a positive correlation was found between the scores in the Psychological Practice dimension and the time spent following a GFD. These results indicate that there is a larger number of factors to analyze than the mere maintenance of a GFD or adherence to it, such as adapting to the lifestyle, becoming comfortable with the activities performed, or becoming familiar with medical protocols. The Management dimension refers to the preparation and knowledge of the patient with respect to prevention, contact with specialists, obtaining information, etc., in light of possible critical conditions and not only the practical development of a GFD.

In the present project, it was observed that the CP obtained higher BMI scores than the NCP, with having celiac disease being a significant factor. These results are in agreement with those from previous studies conducted, most of which were conducted with adults [13,39,41]. We also observed significant inverse relationships between the scores in the Physical Practice dimension and BMI. That is, the lower the score in Physical Practice (healthy diet, physical exercise, adequate rest), the higher the BMI score. These results point to the need to work on the CP beyond the simple adherence to a GFD, seeking to improve their health in a generalized manner, so that these youth do not exhibit cardiovascular diseases later in life, and even to avoid the danger of suffering other metabolic illnesses associated to celiac disease [11,12,70] even if a GFD is strictly followed, as, for example, non-alcoholic fatty liver disease, where these patients have a three-times higher risk than the CD population; as shown in previous studies, a tailored approach to the patient should help to achieve healthy patterns [71].

In summary, the following conclusions were found in the present work: firstly, there are differences in self-care practices between the CP and NCP, mainly in the Physical Practices, Social Practices, and Management dimensions.

Secondly, work guidelines must be established with the CP beyond the adherence to a GFD, seeking an optimal state of health in all the dimensions (physical, psychological, social, and disease management). The CP, despite developing adequate management practices of their illness, showed higher BMI values than the NCP, and lower physical practices (hygiene, rest, sports, and healthy habits) Therefore, health interventions should be aimed at discovering and improving the underlying causes of these young people’s self-care practices.

Thirdly, it is important to base nutrient consumption, within the context of a GFD, on foods that are naturally free of gluten (avoiding processed foods), including enough plant-based foods to increase the consumption of fiber and to achieve a positive balance in the provision of nutrients, to decrease the calorie load.

Fourth, increasing the gluten-free food offerings in restaurant services would allow the CP to be able to attend a greater number of social events without worrying about not following a GFD, and this could benefit their social practices.

And fifth, in general, the present study shows the existing complication of young people with celiac disease when trying to balance the development of their (non-harmful) practices, and to enjoy social and psychological health according to their age [72,73,74]. Nevertheless, we believe it is necessary to continue research on this topic, by obtaining data from the biochemical and quantitative points of view, as well as the points of view of the adolescents and families themselves, with a qualitative perspective, to be able to create effective and efficient frameworks of action. In this way, the research conducted paves the way to continue with the inquiry.

Nevertheless, there are various limitations that must be underlined: first, the data came from a self-completed questionnaire completed in an online platform, and the physiological data were provided by the patients and the families themselves. For this reason, in order to continue research on this matter, a new work protocol has been developed through which we can obtain clinical data from institutional databases. Second, we have only investigated cases of CD without associated pathologies in order to obtain more specific information regarding practices only related to CD. However, for future research, it is recommended that a subgroup be created to obtain information on PC with associated pathologies.

Thirdly, emotional and motivational factors and patient interest can be decisive in their results on self-care and adherence to treatment. This aspect has not been addressed in the present study because it was focused on providing descriptive information on the novel situation. However, on observing the information obtained, we consider it essential to continue working along these lines in the future.

And fourth, the search for the sample was not directly performed through celiac disease associations (seeking to avoid the bias of experience and training), but on the other hand, the sample size could have been greater. However, it is also important to highlight the possible selection bias of the sample, since it was obtained from the centers that volunteered to participate in the reference project.

However, although similar studies describe it as acceptable [75], we propose to continue to try to obtain a larger sample to obtain more reliable results.

## Figures and Tables

**Figure 1 healthcare-12-01796-f001:**
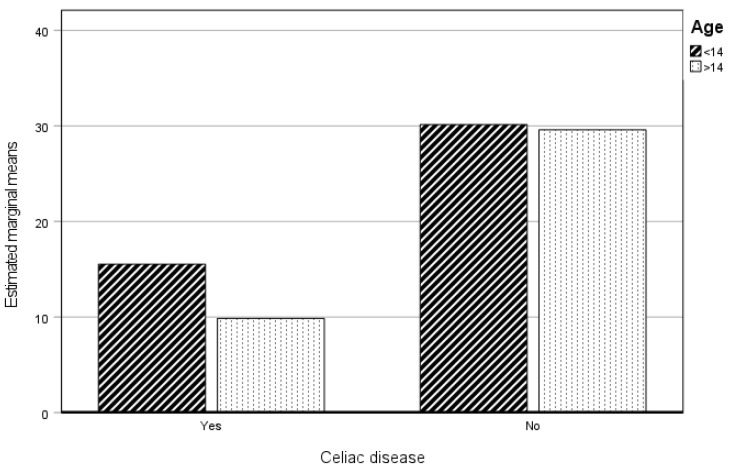
Estimated marginal means of physical practice. CP and NCP.

**Figure 2 healthcare-12-01796-f002:**
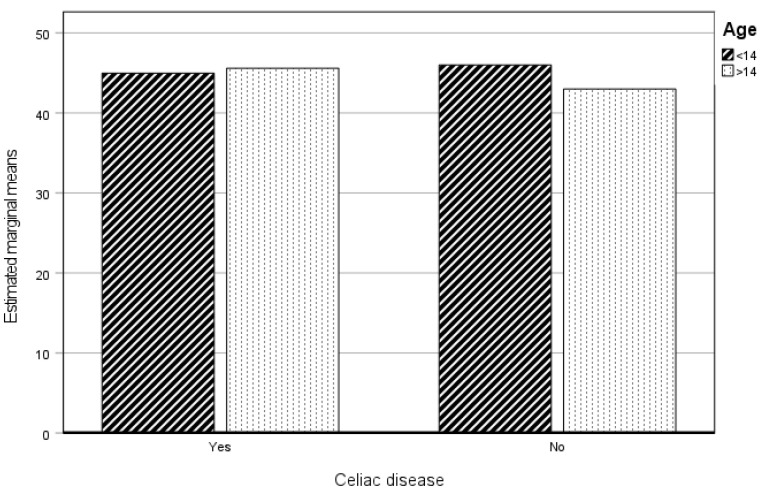
Estimated marginal means of Psychological Practice. CP and NCP.

**Figure 3 healthcare-12-01796-f003:**
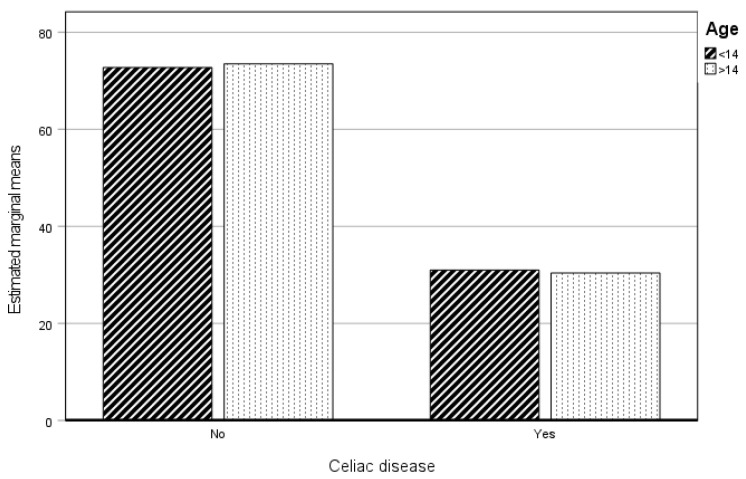
Estimated marginal means of Social Practice. CP and NCP.

**Figure 4 healthcare-12-01796-f004:**
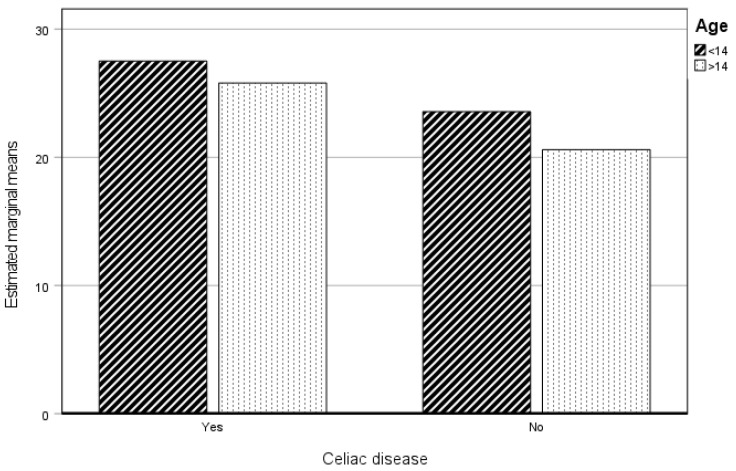
Estimated marginal means of Management. CP and NCP.

**Figure 5 healthcare-12-01796-f005:**
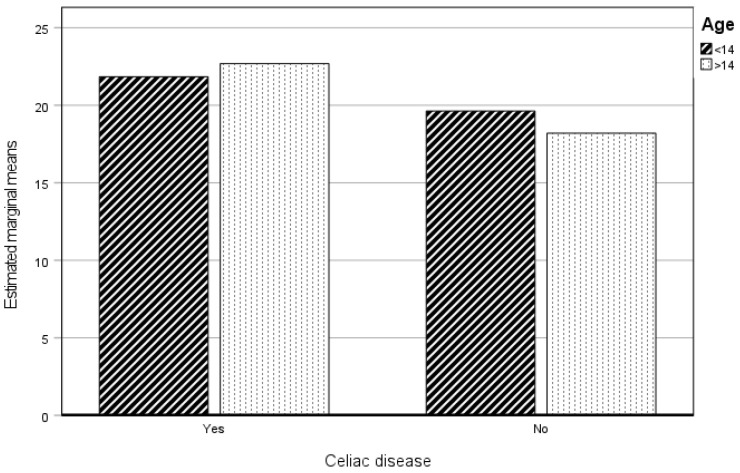
Estimated marginal means of BMI. CP and NCP.

**Table 1 healthcare-12-01796-t001:** Descriptive and correlational analysis of the variables analyzed in the CP.

	Mean (SD)	Asymmetry	Kurtosis	2	3	4	5	6	7
1. Physical (PGJ-C)	11.73 (8.73)	−0.06	−0.11	−0.28 **	−0.02	0.24 **	−0.24 **	0.24 **	−0.10
2. Social (PGJ-C)	30.75 (9.04)	−0.92	0.30		0.20 *	0.45 **	0.10	0.11	0.14
3. Psychological (PGJ-C)	44.75 (7.03)	0.74	10.04			0.15	0.01	0.03	0.18 *
4. Management (PGJ-Q)	26.36 (6.19)	−0.51	−0.34				−0.06	0.30 **	0.03
5 BMI	21.98 (3.81)	0.74	0.73					−0.06	0.03
6. CDA-T	12.37 (7.48)	−0.11	−0.41						−0.06
7. T-GFD	10.85 (3.98)	−0.47	−0.75						

* The correlation is significant at 0.05; ** The correlation is significant at 0.01.

**Table 2 healthcare-12-01796-t002:** Independent sample tests of the CP and NCP group regarding the dimensions of the self-care questionnaire and BMI.

		Levene’s Test for Equality of Variance						
		F	Sig.	T	gl	Significance		Mean Difference
						Single Factor *p*	Two-Factor *p* de	
P. Physical	Equal variances are assumed	3.493	0.264	7.067	235	<0.001	<0.001	18.17938
Manage.	Equal variances are assumed	1.583	0.211	2.178	235	0.016	0.031	4.03107
P. Social	Equal variances are assumed	2.287	0.233	10.372	235	<0.001	<0.001	42.49576
P. Psycho.	Equal variances are assumed	0.002	0.968	0.280	235	0.390	0.780	0.63983
BMI	Equal variances are assumed	7.478	0.107	3.526	235	<0.001	<0.001	1.00533

**Table 3 healthcare-12-01796-t003:** Mean (SD) of the dimensions of self-care and BMI. Comparison between CP and NCP and age.

Celiac Disease	Age	P-Physical	P-Psychological	P-Social	Management	BMI
No	<14	30.14 (6.49)	44.97 (5.51)	72.74 (11.21)	23.57 (6.39)	19.62 (2.09)
	>14	29.60 (3.28)	45.59 (8.45)	73.49 (15.06)	20.60 (1.81)	18.20 (1.06)
	Total	29.91 (5.19)	45.38 (7.58)	73.24 (13.86)	22.33 (5.087)	18.86 (1.72)
Yes	<14	15.53 (7.28)	46.00 (6.45)	31.00 (10.21)	27.51 (5.26)	21.84 (3.92)
	>14	9.86 (8.82)	43.00 (8.18)	30.40 (8.26)	25.79 (6.56)	22.69 (3.78)
	Total	11.73 (8.73)	44.75 (7.03)	30.75 (9.04)	26.36 (6.19)	22.41 (3.83)
Total	<14	17.76 (8.86)	45.13 (5.60)	66.39 (18.70)	26.91 (5.56)	21.50(3.77)
	>14	11.03 (9.78)	45.44 (8.41)	70.92 (17.94)	25.48 (6.49)	22.27 (3.84)
	Total	13.41 (9.97)	45.33 (7.51)	69.32 (18.27)	25.99 (6.19)	22.01 (3.82)

**Table 4 healthcare-12-01796-t004:** Intersubject effects test. Physical Practice, as a dependent variable.

Origin	Type III Sum of Squares	Gl	Mean Square	F	Sig.	Partial Eta Squared
Corrected model	4442.351 ^a^	3	1480.784	22.256	<0.001	0.346
Intersection	19,018.895	1	19,018.895	285.855	<0.001	0.694
Celiac disease	3094.492	1	3094.492	46.510	<0.001	0.270
Age	101.521	1	101.521	1.526	0.219	0.012
Celiac D. * Age	69.175	1	69.175	1.040	0.310	0.008
Error	8383.218	126	66.533			
Total	36,222.000	130				
Corrected total	12,825.569	129				

^a^ R^2^ = 0.346 (adjusted R^2^ = 0.331). Levene’s test for equality of variance (based on the mean = 1.995; gl1 = 3; gl2 = 126; sig = 0.118).

**Table 5 healthcare-12-01796-t005:** Intersubject effects test. Psychological Practice as a dependent variable.

Origin	Type III Sum of Squares	Gl	Mean Square	F	Sig.	Partial Eta Squared
Corrected model	40.765 ^a^	3	13.588	0.236	0.871	0.006
Intersection	84,598.166	1	84,598.166	1471.067	<0.001	0.921
Celiac disease	6.461	1	6.461	0.112	0.738	0.001
Age	14.854	1	14.854	0.258	0.612	0.002
Celiac D. * Age	34.392	1	34.392	0.598	0.441	0.005
Error	7246.012	126	57.508			
Total	274,421.000	130				
Corrected total	7286.777	129				

^a^ R^2^ = 0.006 (adjusted R^2^ = −0.018). Levene’s test (based on the mean = 1.429; gl1 = 3; gl2 = 126; sig = 0.238).

**Table 6 healthcare-12-01796-t006:** Intersubject effect test. Social Practice as the dependent variable.

Origin	Type III Sum of Squares	Gl	Mean Square	F	Sig.	Partial Eta Squared
Corrected model	19,686.048 ^a^	3	6562.016	35.348	<0.001	0.457
Intersection	113,111.658	1	113,111.658	609.313	<0.001	0.829
Celiac disease	18,882.959	1	18,882.959	101.719	<0.001	0.447
Age	0.059	1	0.059	0.000	0.986	0.000
Celiac D. * Age	4.782	1	4.782	0.026	0.873	0.000
Error	23,390.383	126	185.638			
Total	667,816.000	130				
Corrected total	43,076.431	129				

^a^ R^2^= 0.457 (adjusted R^2^ = 0.444). Levene’s test (based on the mean = 2.464; gl1 = 2; gl2 = 126; sig = 0.065).

**Table 7 healthcare-12-01796-t007:** Intersubject effect test. Management as the dependent variable.

Origin	Type III Sum of Squares	Gl	Mean Square	F	Sig.	Partial Eta Squared
Corrected model	279.575 ^a^	3	93.192	2.513	0.062	0.056
Intersection	24,931.197	1	24,931.197	672.170	<0.001	0.842
Celiac disease	219.119	1	219.119	5.908	0.016	0.045
Age	57.630	1	57.630	1.554	0.215	0.012
Celiac D. * Age	4.139	1	4.139	0.112	0.739	0.001
Error	4673.417	126	37.091			
Total	92,781.000	130				
Corrected total	4952.992	129				

^a^ R^2^ = 0.056 (adjusted R^2^ = 0.034). Levene’s test (based on the mean = 4.028; gl1 = 3; gl2 = 126; sig = 0.009).

**Table 8 healthcare-12-01796-t008:** Intersubject effect test. BMI as the dependent variable.

Origin	Type III Sum of Squares	Gl	Mean Square	F	Sig.	Partial Eta Squared
Corrected model	6.037 ^a^	3	2.012	2.554	0.058	0.056
Intersection	129.029	1	129.029	163.775	<0.001	0.559
Celiac disease	4.533	1	4.533	5.753	0.018	0.043
Age	0.085	1	0.085	0.108	0.742	0.001
Celiac D. * Age	0.648	1	0.648	0.822	0.366	0.006
Error	101.632	129	0.788			
Total	548.000	133				
Corrected total	107.669	132				

^a^ R^2^ = 0.056 (adjusted R^2^ = 0.034). Levene’s test (based on the mean = 12.934; gl1 = 3; gl2 = 126; sig < 0.001).

## Data Availability

Data are available by contact with the correspondence authors.

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
