# Peer review of "Comparison of Self-Care Practices and BMI between Celiac and Non-Celiac Adolescent Populations"

_healthcare, 2024, doi:10.3390/healthcare12171796_

Round 1

Reviewer 1 Report

Comments and Suggestions for Authors

The study addresses an important topic in comparing self-care practices and BMI between adolescents with and without celiac disease. Overall, the manuscript has several strengths but could benefit from some revisions to improve clarity and impact. My main suggestions are:

Introduction:

1.     Provide more background on the prevalence and impact of celiac disease specifically in adolescents, as this is the focus population.

2.     Include a clearer rationale for why comparing self-care practices between celiac and non-celiac adolescents is important.

3.     State the specific research questions/hypotheses more explicitly at the end of the introduction.

Discussion:

1.     Expand on the interpretation of key findings, especially the differences observed between celiac and non-celiac groups in physical practices, social practices, and management.

2.     Discuss potential mechanisms underlying the association between GFD duration and higher BMI.

3.     Address limitations of the study more thoroughly, including potential selection bias in the sample.

4.     Provide clearer implications for clinical practice and future research directions.

Author Response

Thank you very much for your review. Below we answer your recommendations one by one, indicating the place where they have been inserted or modified in the text.

Introduction:

  1. Provide more background on the prevalence and impact of celiac disease specifically in adolescents, as this is the focus population.

Added reference to prevalence in minors (line 32-34).

Added implications for children and youth (line 38-40).

  1. Include a clearer rationale for why comparing self-care practices between celiac and non-celiac adolescents is important.

Including. Line 54-57

  1. State the specific research questions/hypotheses more explicitly at the end of the introduction.

Including. Line 100-107

Discussion:

  1. Expand on the interpretation of key findings, especially the differences observed between celiac and non-celiac groups in physical practices, social practices, and management.

In general, clarifications have been made in the sentences referring to the interpretations of the results in order not to be so generalized. In addition, a reference bibliography has been added.

In addition, information has been added in the following lines on the contents: questionnaire (320-322), 332, 334, social (345-350), psychological (360-365) (368-370), physical practices (382-387).

  1. Discuss potential mechanisms underlying the association between GFD duration and higher BMI.

In our study we found no significant relationship between time with GFD and BMI. the sentence in the abstact that gave erroneous information has been modified. We only found a significant relationship between BMC and physical practice. The discussion of these results can be seen in lines 373 to 386.

  1. Address limitations of the study more thoroughly, including potential selection bias in the sample.

In accordance with their recommendations, the limitations have been further discussed and the sample selection bias has been addressed. Linens 416-421 and 424-426.

  1. Provide clearer implications for clinical practice and future research directions.

Added in the text as suggested. Lines 320-322; 394-396; 419-421.

Reviewer 2 Report

Comments and Suggestions for Authors

In this the study, the authors aimed to provide information on the situation of adolescent celiac disease (CD) patients with respect to their process of self-care and BMI parameters during the gluten-free diet (GFD) regimen, by comparing them to a population of adolescents without CD. A total of 118 adolescents with CD participated in the study. The recruitment of the non-celiac disease group as control group was performed through the randomized sampling of 1280 surveys of participants to obtain 118 participants for the group.

They found that the CD population analyzed showed significant correlations between the physical practices and management with adherence to a GFD, and between a GFD and psychological practices. There were differences between the celiac and the non-celiac populations in the dimensions physical
practices, social practices, and management. Importantly, with respect to BMI, age and CD showed a significant impact (p<.001). The length of time following a GFD was associated to a higher BMI (p<.001). They concluded that the application of multi-dimensional questionnaires and their relationship with the adherence to a GFD provide valuable information to improve the management of this population.

The study is of interest and of potential clinical relevance. However, some issues require additional informations and should be addressed.

-Enrrolled CD patients: the authors should report the median age of CD diagnosis as well as the presence (if any) of comorbidities such as thyroid disorders or diabetes or other, potentially impacting the metabolic profile of patients and measured variables.

-GFD adherence: in the discussion, the authors discuss the problem for measuring the level of adherence to a GFD. In this regard, they should recall previous studies demonstrating that serum IgA antibodies such as IgA anti-actin antibodies, a marker of severe mucosal damage, and IgA Anti-Saccharomyces cerevisiae, a marker of increased mucosal permeability, disappear in CD patients who well follow GFD, as previously demonstrated (Clin Exp Immunol. 2004 Aug;137(2):386-92. doi: 10.1111/j.1365-2249.2004.02541.x.; Aliment Pharmacol Ther. 2005 Apr 1;21(7):881-7. doi: 10.1111/j.1365-2036.2005.02417.x.)

-Discussion: the authors should recall and discuss their results with previous studies demonstrating that more than one-third of CD patients adhering to a GFD had concurrent non-alcoholic fatty liver disease (NAFLD), beyond traditional metabolic factors, accounting for a three-fold increased risk compared to the general population suggesting that dietary advice provided using a patient-tailored approach should assist CD patients with NAFLD in achieving an appropriate nutritional intake whilst reducing the risk of long-term liver-related events, as previously demonstrated (Aliment Pharmacol Ther. 2018 Sep;48(5):538-546. doi: 10.1111/apt.14910.)

Author Response

Thank you very much for your contributions. We will now respond to your comments by referring to the changes or modifications made to the text.

-Enrrolled CD patients: the authors should report the median age of CD diagnosis as well as the presence (if any) of comorbidities such as thyroid disorders or diabetes or other, potentially impacting the metabolic profile of patients and measured variables.

The mean age at diagnosis was included in line 202. The time developing GFD coincided with the time of diagnosis.

As inclusion criteria we indicated that there was no diagnosis of chronic metabolic or cardiovascular disease (associated or not with CD), in order to clearly observe the practices derived from the development of GFD without interference from another disease. Line 135

-GFD adherence: in the discussion, the authors discuss the problem for measuring the level of adherence to a GFD. In this regard, they should recall previous studies demonstrating that serum IgA antibodies such as IgA anti-actin antibodies, a marker of severe mucosal damage, and IgA Anti-Saccharomyces cerevisiae, a marker of increased mucosal permeability, disappear in CD patients who well follow GFD, as previously demonstrated (Clin Exp Immunol. 2004 Aug;137(2):386-92. doi: 10.1111/j.1365-2249.2004.02541.x.; Aliment Pharmacol Ther. 2005 Apr 1;21(7):881-7. doi: 10.1111/j.1365-2036.2005.02417.x.)

Suggested literature has been added to the other reference literature in the text. Line 326.

-Discussion: the authors should recall and discuss their results with previous studies demonstrating that more than one-third of CD patients adhering to a GFD had concurrent non-alcoholic fatty liver disease (NAFLD), beyond traditional metabolic factors, accounting for a three-fold increased risk compared to the general population suggesting that dietary advice provided using a patient-tailored approach should assist CD patients with NAFLD in achieving an appropriate nutritional intake whilst reducing the risk of long-term liver-related events, as previously demonstrated (Aliment Pharmacol Ther. 2018 Sep;48(5):538-546. doi: 10.1111/apt.14910.)

Suggested discussion has been added. Line 362.

Round 2

Reviewer 2 Report

Comments and Suggestions for Authors

The revised manuscript satisfactorily addressed the raised points and can be now accepted.